# Facile Green Preparation of Reduced Graphene Oxide Using Citrus Limetta-Decorated rGO/TiO₂ Nanostructures for Glucose Sensing

Medha Gijare [1], Sharmila Chaudhari [2], Satish Ekar [1], Shoyebmohamad F. Shaikh [3], Rajaram S. Mane [4,*], Bidhan Pandit [5], Muhammad Usman Hassan Siddiqui [6] and Anil Garje [7,*]

1   Department of Physics, Baburaoji Gholap College, Pune 411027, India
2   Department of Physics, Anantrao Pawar College, Pirangut, Pune 412115, India
3   Department of Chemistry, College of Science, King Saud University, P.O. Box 2455, Riyadh 11451, Saudi Arabia
4   School of Physical Sciences, Swami Ramanand Teerth Marathwada University, Nanded 431606, India
5   Department of Materials Science and Engineering and Chemical Engineering, Universidad Carlos III de Madrid, Avenida de la Universidad 30, Leganés, 28911 Madrid, Spain
6   Unilever Lipton Jebal Ali Factory Engineering Department, Dubai P.O. Box 325, United Arab Emirates
7   Department of Physics, Sir Parashurambhau College, Pune 411030, India
\*   Correspondence: rajarammane70@gmail.com (R.S.M.); anil.garje@spcollegepune.ac.in (A.G.)

**Abstract:** The important electrochemical measurements of reduced graphene oxide-titanium oxide (rGO)/TiO₂) electrodes for the application of a glucose sensor are reported in the proposed work. Investigating the sensitivity, stability, and reproducibility of sensor electrodes that were made and used to evaluate the concentration of glucose in the serum is one of the novel aspects of this work. This study presents the use of citrus limetta (sweet lime) fruit peel waste to synthesize a green reduction of graphene oxide (rGO). The rGO/TiO₂ composite obtained using the microwave heating method is applied for measuring the structural and morphological properties by various means. A conducting fluorine-tin oxide substrate is used to modify the enzymeless glucose sensor electrode. The electrochemical measurements of rGO/TiO₂ sensor electrodes are carried out using the technique of cyclic voltammetry. The rGO/TiO₂ sensor electrode exhibits a high sensitivity of 1425 $\mu A/mM\,cm^2$ towards glucose concentration in the range of 0.1 to 12 mM. The sensor was found to be extremely stable and repeatable with a response time of 5 s along with a minimum detection limit of 0.32 $\mu M$ of glucose. The rGO/TiO₂ sensor shows relative standard deviation (RSD) of 1.14%, 1.34%, and 1.3% which reveals its excellent stability, repeatability, and reproducibility respectively. The sensor was used for glucose level detection in natural blood serum and shows an RSD of 1.88%. which is in good agreement with the commercial glucose sensor values.

**Keywords:** cyclic voltammetry; glucose sensor; graphene oxide; microwave heating; phytochemicals

## 1. Introduction

According to the International Diabetes Federation, the metabolic disorder known as diabetes has become a major issue for people around the world, affecting millions of people. This epidemic threatens human health and is one of the most significant contemporary barriers to long-term social development and economic growth. The ability to accurately monitor and control individuals' blood glucose levels is critical for diabetic patients to receive the proper diagnosis and treatment. To avoid diabetes emergencies like hypoglycemia (low blood sugar, ≤3 mM), the physiological glucose level must be checked on a routine basis [1]. Glucose detection has become increasingly important in the fight against diabetes and in preventing financial loss due to the significant financial cost caused by diabetes and its serious complications. As a result, for a few decades, developing reliable and low-cost glucose sensors with excellent sensitivity and selectivity has had enormous social and

economic importance. Glucose biosensors have been significantly subsidized to monitor glucose levels in diabetic patients [2]. Many researchers are actively engaged in developing new biosensors for various applications in various disciplines since Clark and Lyons first proposed the concept of enzymatic electrodes in 1962. Enzymatic sensors confirmed good sensitivity and stability but demonstrated a few drawbacks like poor reproducibility, complex mobility, etc. [3,4]. Though enzymatic sensors have certain advantages, commercial applications are mitigated on account of their high manufacturing cost [5]. Non-enzymatic glucose sensors are a solution created in response to these limitations [6].

All candidate materials for the electrocatalyst in glucose detection, including metal oxide, carbon materials, polymers, graphene metal, graphene oxide materials, etc., have distinguished themselves [7]. Graphene is a two-dimensional material made up of a single layer of carbon atoms in a hexagonal lattice. It has received considerable attention due to its unusual physical, optical and electrical properties. It is the world's thinnest and toughest material, with exceptional electrical and thermal conductivity. Graphene is synthesized using a variety of methods, including micromechanical cleavage, chemical vapor deposition, epitaxial growth on silicon carbide liquid-phase exfoliation, graphene oxide reduction by various reducing agents, and graphite intercalation [8]. It is astonishing mechanical [9], thermal [10] and electrical [11] properties have captivated tremendous attentiveness. Reducers such as hydrazine or sodium borohydride are typically envisaged during the chemical preparation of GO [12,13]. These reducing agents have some instinctive disadvantages such as expensive and highly toxic. Moreover, the Van der Waals forces among the graphene layers produce an irreversible aggregation of GO [14].

In the case of green reduction, a study of suitable phytochemicals as reducing agents is necessary. Additionally, a green reduction is a simple, eco-friendly, and cost-effective method. In light of this motivation, the work that is being proposed provides an environmentally friendly reduction of GO using Citrus limetta (sweet lime/Mosambi fruit waste) and $TiO_2$ electrodes for electrochemical glucose sensing that does not require enzymes. Green reduction and bio-synthesis routes are employed to overcome the adverse effects of hazardous substances, where phytoextracts or bacteria microorganisms play eco-friendly (for reduction) and capping agent roles [15–17]. A germ-free environment is necessary to grow micro-organisms, and also expertise is needed in this process. The reduction of GO can be done using citrus plant extract, which has a slower reaction rate to control the crystal growth. In the past, the green synthesis of reduced graphene oxide (rGO) has been studied using several plants and fruit extracts [18]. The green nature, low cost, and easy local availability of plant extracts and fruit wastes make them suitable for reduced graphene oxide. Citrus limetta is the primary natural source of vitamin C and an energy booster [19]. Fruit peels of sweet lime are a waste to the juice industry, an important source of phytochemicals like flavonoids, pectins, carotenoids, hesperidin, terpineol alkaloids, essential oils, *d*-limonene, and extraordinary antioxidants as well as antibacterial properties [20]. Graphene obtained via the green reduction process can be functionalized with metal oxides for various applications including electrochemical supercapacitors, batteries, fuel cells, magnetic data storage, water splitting, chemical sensors, water filters, dye removal, and dye-sensitized solar cells, etc. [21–24].

In the present study, citrus limetta was chosen as a reducing agent for the synthesis of rGO because it has a small effect on the environment and doesn't have any harmful biological reactions that could happen. The soxhlet extraction method is preferred due to its simplicity and effectiveness in collecting sweet lime peel extract. Functionalized graphene with some metallic oxide revealing extraordinary sensing properties is familiar for a long [25]. The electrochemical deposition method adopted by Luo et al. to produce graphene and CuO nanocubes [26]. The rGO obtained by Yang et al. using the chemical reduction method demonstrated excellent sensitivity and selectivity of 4 mM and 7.5 mM linear ranges y [27]. Komathi et al. presented the development of electrochemical biosensors by functionalizing titanium dioxide ($TiO_2$) nanowires/graphene as nano-stocks [28]. According to these findings, the graphene-based metal/metal oxide nanoparticles endow

good sensitivity, a wide linear range, and a low detection limit with good selectivity. Hence, the $rGO/TiO_2$ composite has been selected as a sensor electrode in the proposed work and used as an enzyme-free biosensor to detect glucose concentration efficiently using a cyclic voltammetry (CV) test. Moreover, very little literature on green synthesis of the $rGO/TiO_2$-based biosensors is available is familiar. The as-fabricated glucose biosensor is also used to investigate glucose levels in the existence of obstructing substances in real blood serum.

## 2. Materials and Methods

### 2.1. Materials

All the chemicals used were of the analytical grade, meaning they were of the highest purity. Graphite powder (98%), $KMnO_4$ (99%), $NaNO_3$ (98%), $H_2O_2$ (30%), $H_2SO_4$ (99.99%), HCl (30%), $H_3PO_4$ (85%), KCl (99.99%) and NaCl (99%) $TiO_2$ (99.9%), etc., were obtained from Finar chemicals (India). Qualigens fine chemicals in India provided polyvinyl alcohol (PVA, 99%), ethanol, Fluorine-doped Tin Oxide (FTO) substrates, D (+) glucose, dopamine, L-ascorbic acid, D (-) fructose, and lactose, phosphate buffer solution (PBS) was prepared according to AAT Bioquest, Inc. [29] and sweet limes were bought from a local market.

### 2.2. Preparation of GO

A modified Hummer method was used to synthesize GO [30]. Briefly, the concentrated $H_2SO_4$ and $H_3PO_4$ were mixed in a 9:1 proportion in a round bottom flask, followed by graphite powder and $NaNO_3$ in a 2:1 proportion. To achieve the desired result, it was necessary to stir the mixture at 100 °C continuously for an hour before adding the $KMnO_4$. Adding $H_2O_2$ to the mixture served the purpose of oxidation, resulting in the formation of a yellow-colored product. $SO_4^{2-}$ is completely removed from the sample by repeatedly washing with 5% HCl and deionized (DI) water, followed by centrifugation (4000 rpm). After nearly 24 h of drying at 60 °C, the resultant product (GO) was obtained which was collected for the next processes.

### 2.3. Preparation of Aqueous Phytoextract

Fresh sweet limes were purchased from a local market and cleaned using DI water. The citrus limetta fruit waste, such as peel, was cut into small pieces with a domestic cutter. In soxhlet extraction, 15 gm of finely cut peels were placed in a thimble of soxhlet tube, and the DI water was used as the solvent. During condensation, solutes are moved to the reservoir for repeated cycles. The system was cooled for 10–12 h and then the collected extract was preserved at 4 °C for further use.

### 2.4. Mechanism of Reduction of GO by Citrus Limetta Phytoextract

Citrus Limetta phytoextract can reduce GO by utilizing the oxidized products found in sweet lime fruit peel. These oxidized products include flavonoids, polyphenols, tannis, terpenoids, and saponins, among others. These phytochemicals show high binding linkage towards $O_2$- containing functionalities to produce their respective oxides and water molecules. Flavonoid compounds are reduced to the phenolic hydroxyl groups. While reacting with the $O_2$- containing groups, the phenolic hydroxyl group may release water molecules on GO sheets. GO contains reactive, hydroxyl, epoxide, and reactive carbonyl species. As shown in Figure 1, acidic polyphenols in the aqueous GO and Citrus Limetta phytoextract were combined, ultra-sonicated, and refluxed at 95 °C. Through a nucleophilic reaction, epoxide fraction and phytoextracts could react with each other. This caused the carbon-oxygen bond to break. Carbonyl and hydroxyl groups in polyphenols could be attacked by nucleophiles in the same way, which would destroy water molecules. GO is changed into rGO in this process, which makes graphene, which is a stable black compound.

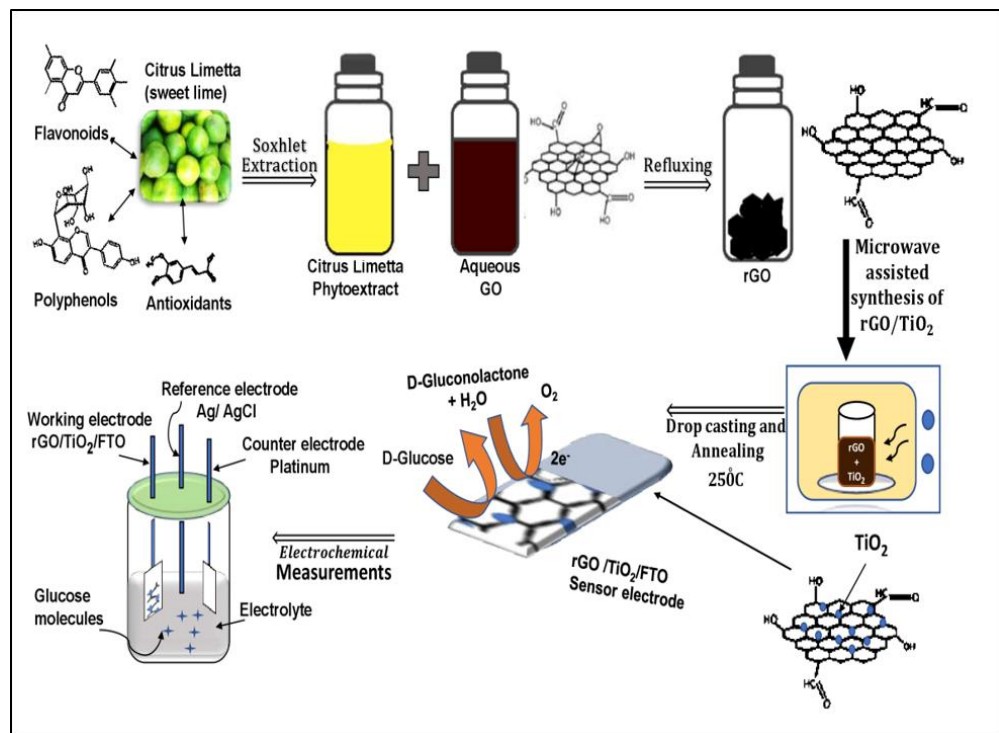

**Figure 1.** Mechanism of green reduction of GO and rGO/TiO$_2$/FTO for glucose sensing application.

### 2.5. Reduction of GO Using Aqueous Citrus Limetta Peel Extract

Briefly, 100 mg GO (0.1 mg/mL) was disseminated in ultrapure water by probe sonication for 40 min. The GO solution was mixed with 5 wt.% of 10 mL of aqueous sweet lime fruit extract. The solution's composition was refluxed at 98 °C for 8 h until it turned brown-black. The supernatant (rGO) was repetitively washed with DI water which was centrifuged at 4500 rpm, followed by drying at 100 °C in the oven.

### 2.6. Synthesis of rGO/TiO$_2$ Nanocomposite

The rGO/TiO$_2$ composite was prepared using microwave treatment. A fine powder of rGO and TiO$_2$ in a 2:1 ratio was dispersed in DI water. After sonication, the mixture was stirred. The compound was kept at 300 °C for 15 min in a microwave (500 W). Finally, the rGO/TiO$_2$ nanocomposite dry powder was collected. First, 10 mg of rGO/TiO$_2$ nanocomposites were mixed with 5 mL of DI water and 5 µL of PVA and sonicated in a bath for 30 min. Then, 10 µL of the mixture was drop-cast onto a clean conducting FTO substrate. The working area of the electrode was 1 cm$^2$. The as-prepared electrode was annealed to remove the binder at a temperature of 250 °C for 1 h. A three-electrode setup consisting of a working electrode made of rGO/TiO$_2$/FTO, a reference electrode made of Ag/AgCl, and a counter electrode made of platinum was used to conduct electrochemical measurements for glucose detection.

### 2.7. Glucose Sensing through rGO/TiO$_2$

For rGO/TiO$_2$ nanocomposite preparation, in-situ thermal reductions of rGO and TiO$_2$ were developed. The epitaxial growth process was used to prepare TiO$_2$ nanobelts. Using Phosphate-Buffered Saline (PBS) as an electrolyte, the glucose sensing performance of the rGO/TiO$_2$/FTO modified electrode was evaluated. When glucose dissolves in PBS, D-gluconolactone and hydrogen peroxide are produced (H$_2$O$_2$). Electro-oxidation of glucose at TiO$_2$ grain boundaries also produces D-gluconate with H$^+$ ions. As a result, gluconolactone is the primary product of the oxidation process that hydrolyzes gluconic

acid. By releasing electrons, lowering the barrier potential between successive grains, and increasing electrical conductivity, $H^+$ ions reduce pre-absorbed oxygen [31].

$$\text{Glucose} + O_2 \rightarrow H_2O_2 + \text{gluconic acid} \tag{1}$$

*2.8. Characterization Techniques*

The as-synthesized GO, rGO, and rGO/TiO$_2$ sample products were analyzed using a Bruker D8-Advanced Diffractometer with Cu Kα radiation (λ = 0.154 nm) from 5° to 85° at a scanning rate of 2°/min for structural elucidation. Field Emission Scanning Electron Microscopes (FE-SEM): FEI Nova Nano SEM 450 was used to examine the surface morphology and elemental composition of GO, rGO, and rGO/TiO$_2$ samples. A Micro-Raman spectrometer (Jobin-Yuon HR 800 UV) with a He–Ne (633 nm) laser excitation source was used to conduct the Raman analysis. The multifunctional X-ray photoelectron spectroscopy (XPS) was used to perform XPS analysis on GO and rGO (PHI ulvac probe III Scanning Microprobe). The Fourier-transform infrared spectroscopy (FTIR) study was conducted using an FTIR-6100 spectrometer (JASCO) in transmission mode in the wavenumber range 4000–400 cm$^{-1}$.

## 3. Results and Discussion

### 3.1. Structure Elucidation Confirmation

The crystal structures of the GO, rGO, and rGO/TiO$_2$ were characterized and confirmed using X-ray diffraction spectra, respectively. As shown in Figure 2a, the reflection peak obtained at 11.8° indicates the presence of oxygen functionalities on the GO surface, which has an interlayer basal spacing of 7.4 Å, suggesting the presence of oxygen functionalities on the GO surface (0 0 2). The spacing between the graphite grains was 3.36 Å. There could be several oxygen-containing groups on the edges of each layer in the first one, which eventually would have increased the distance between them. A broad peak around 25.17° was assigned to the (0 0 2) basal plane of rGO with a separation of 0.37 nm. The fall in basal space and broad peak at 24.7° suggested that GO was transformed to rGO. The prior XRD patterns of rGO/TiO$_2$ confirmed peaks at 25.3°, 37.8°, 48.0°, 53.8°, 55.0°, 62.6°, and 68.8°, for the (1 0 1), (0 0 4), (2 0 0), (1, 0 5), (2 1 1), (2 0 4), and (1 1 6) reflection planes), respectively of anatase TiO$_2$ (JCPDS#21-1272 [32]. The narrow and sharp peaks confirmed a high degree of crystallinity of rGO/TiO$_2$, supporting an easy electron transfer in an electrochemical reaction. Nevertheless, there were no broad peaks corresponding to rGO in the rGO/TiO$_2$ nanocomposites. The rGO surface can be extensively coated on the TiO$_2$ nanostructures, which has the effect of suppressing XRD signals that are caused by the stacking of rGO layers [33]. The average crystallite size (D) calculated using the Debye-Scherrer formula is as follows,

$$D = (0.9 \times \lambda)/(\beta \times \text{Cos}\theta) \tag{2}$$

where, λ is a wavelength of Cu Kα radiation of 1.54 Å, and β is the full width of the diffraction peak at the half maximum at Bragg diffraction angle 2θ. Table 1 shows the XRD results of the rGO/TiO$_2$ nanocomposite. The average crystallite size for TiO$_2$ was 35 ± 2 nm.

The D-bands of GO (black color), rGO (red color), and rGO/TiO$_2$ (blue color) at 1345 cm$^{-1}$, 1349 cm$^{-1}$, and 1353 cm$^{-1}$ were respectively responsible for the obtained peaks (Figure 2b). They correspond to the structural defects developed by attaching various functional assemblies with carbon. The peaks at 1593 cm$^{-1}$, 1598 cm$^{-1}$, and 1604 cm$^{-1}$ reflect the G band, which appears due to the primitive scattering of the E$_{2g}$ phonon of $sp^2$ carbon atoms in the same experiment. The wide 2D band of GO with a higher wavenumber was found at 2917 cm$^{-1}$, confirming the multilayered nature of GO. The reduction of GO to rGO using citrus limetta phytoextract, which causes rGO sheets to stack, was predicted by the two D bands of rGO. Furthermore, the ID/IG ratio was used to measure structural disorder, which increased slightly from 0.85 to 0.87 as GO was reduced to rGO. The higher

ID/IG ratio could be assigned to higher structural defects. As a result, defects form in addition to removing oxide functional groups attached to the GO surface. Despite this, the ID/IG ratio for the rGO/TiO$_2$ nanocomposite was 0.6. This was assigned to the microwave heating process during the reduction of $sp^3$ carbon to $sp^2$ carbon, causing the $sp^2$ domain size of carbon atoms to become decreased [34]. In rGO/TiO$_2$ spectrum, the peaks below 1000 cm$^{-1}$ may be attributed to the A$_g$ and B$_g$ modes of TiO$_2$. The crystalline size of the rGO/TiO$_2$ nanocomposite was estimated using Tuinstra- Koening relationship as [35,36],

$$L_a = \left(2.4 \times 10^{-4}\right) \times \left(\frac{I_G}{I_D}\right) \times \lambda^4 \tag{3}$$

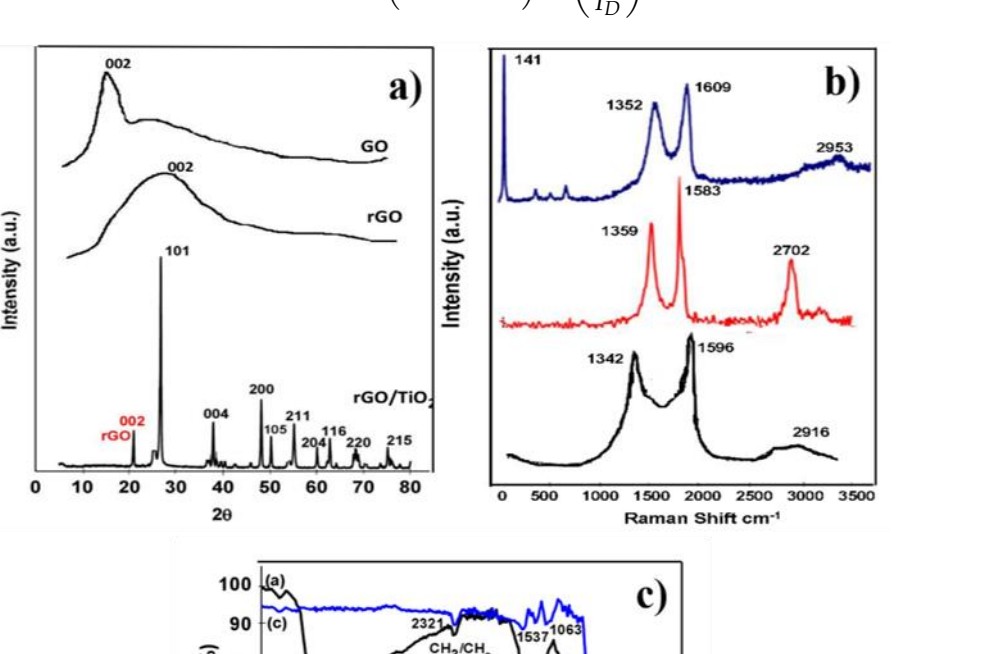

**Figure 2.** (**a**) X-ray diffraction, (**b**) Raman shift, and (**c**) FTIR analysis spectra of GO, rGO, and rGO/TiO$_2$.

**Table 1.** XRD analysis for rGO and TiO$_2$ products.

| No | Two Theta (deg.) | (h k l) | Phase | Crystallite Size (D) $\pm$ 2 (nm) |
|---|---|---|---|---|
| 1 | 20.9 | 002 | rGO | 19 |
| 2 | 26.7 | 101 | TiO$_2$ | 81 |
| 3 | 37.89 | 004 | TiO$_2$ | 14 |
| 4 | 48.14 | 200 | TiO$_2$ | 41 |
| 5 | 49.89 | 105 | TiO$_2$ | 46 |
| 6 | 55.13 | 211 | TiO$_2$ | 41 |
| 7 | 60.05 | 204 | TiO$_2$ | 22 |
| 8 | 62.77 | 116 | TiO$_2$ | 22 |
| 9 | 68.31 | 220 | TiO$_2$ | 23 |
| 10 | 75.15 | 215 | TiO$_2$ | 47 |

$L$ = crystallite size, $\lambda$ = wavelength of the laser source, and $I_D/I_G$ is the Raman peak intensity ratio of $D$ and $G$ bands, respectively. The nanocomposite's crystallite size was found to be $34 \pm 2$ nm, which agrees well with XRD analysis. The $O_2$ functional groups in GO were confirmed by the broad O-H stretching vibrational band, C=O, and epoxy group at 3364 cm$^{-1}$, 1739 cm$^{-1}$, and below 1000 cm$^{-1}$, respectively (Figure 2c). A few absorption peaks at 2340 cm$^{-1}$, 1638 cm$^{-1}$, and 993 cm$^{-1}$ of C-H stretching, bending vibrations of C=C, and aromatic =C-H around, respectively, were obtained. For rGO (curve b of Figure 2c), the peak at 3340 cm$^{-1}$ changed to broad and less intense than GO, revealing the significant removal of the hydroxyl group. Also, it was noticed that the other peaks became less intense compared to the GO at the same location. It could be due to the incomplete removal of $O_2$ functionalities while synthesizing GO using citrus limetta phytoextract. In short, the polar functional groups support hydrophilic properties and high dispersibility of synthesized rGO in DI [37]. The peak noted at 677 cm$^{-1}$ was credited to Ti-O-Ti and Ti-O-C in rGO/TiO$_2$ composite (curve c).

### 3.2. Morphological Study

The FE-SEM surface micrograph of GO (Figure 3a), synthesized using an improved Hummer method, clearly demonstrated a two-dimensional sheet-like structure. As GO has a multiple laminar layer structure the edges of individual sheets are distinguished from one another in the FE-SEM images. The films are stacked one on top of the other, with some wrinkles. As per reported literature [38], the thickness of GO sheet was significantly greater than the thickness of the single-layer graphene. The increase in thickness results from the oxygen-containing functional groups that were introduced. It can also be noted that the GO sheets were thicker at the edges [38]. Since the oxygen-containing functional groups could primarily be combined at the edges of the GO. It is clear from the FE-SEM images that the GO sheets are firmly suspended in their frames. The FE-SEM image of rGO (Figure 3b) obtained from GO demonstrates nanosheets' agglomeration and the irregular arrangement of stacked crumpled sheets. This could be attributed to the nucleophilic reaction of flavonoids and polyphenols, which occurred during the synthesis of rGO and involved the removal of water molecules from carbonyl and hydroxyl groups. The difference in morphological characteristics between GO's leafy, stacked structure and rGO wrinkled structure suggests that the green reduction process has prime importance in GO to rGO conversion.

The granular shape of TiO$_2$ nanoparticles displayed in Figure 3c is attributed to the aggregation of nanoparticles during the evaporation process due to microwave heating [39]. The elemental compositions of rGO and rGO/TiO$_2$ were studied using an energy-dispersive spectrum. Figure 3d depicts carbon (C) and oxygen (O) contents in rGO and titanium (Ti) in rGO/TiO$_2$. From the EDS analysis, it is evident that the rGO contains around 84.88 at. % C, 15.12 at. % O whereas rGO/ TiO$_2$ avails about 65.99 at.% C, 28.95 at.% O and 28.95 at.% Ti, confirming the formation of TiO$_2$ nanostructures on the rGO surface.

### 3.3. X-ray Photoelectron Spectroscopy Analysis

Figure 4a–c displays the XPS spectrum of the C1s spectrum of GO, rGO, and rGO/TiO$_2$. The peak in Figure 4a that corresponds to the C1s bands in deconvoluted GO into C-C, C=O, and C(O)OH bonds, respectively, has binding energies of 284.8, 286.5, and 288.5 eV [40]. When compared to GO, the C=O and C(O)OH bands of the rGO C1s spectrum show a marked decrease in intensity, indicating that rGO has undergone deoxygenation. However, as shown in Figure 4b, after being reduced, the peak oxygen intensity with functionalities is sharply reduced. These results have demonstrated the decrease in GO. The C-C, C=O, and C(O)OH bonds, respectively, are ascribed to the peaks with binding energies of 284.5, 285.4, and 283.7 eV that correspond to the C1s bands in deconvoluted rGO. The variation in the chemical surrounding of carbon atoms after reduction may cause a shift in binding energy [41]. Additionally, the surface C/O atomic ratios increased from 2.12 to 5.57, validating the successful removal of $O_2$ groups from the GO surface using aqueous

peel extract of citrus limetta. In Figure 4c, three peaks found in the C1s deconvoluted spectrum of rGO/TiO$_2$ are for C-C/C-H $sp^3$ hybridized carbon bonds, C-O bonds, and the $sp^2$ carbon bonds of graphene (c), respectively.

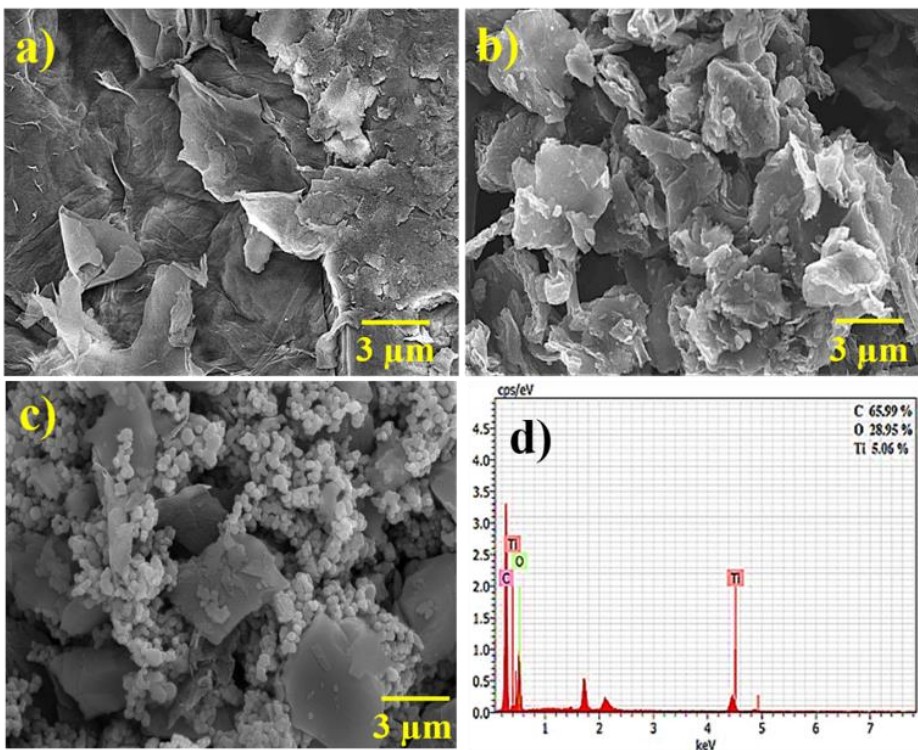

**Figure 3.** FE-SEM images of; (**a**) GO, (**b**) rGO, (**c**) rGO/TiO$_2$, and (**d**) EDS spectrum of rGO/TiO$_2$.

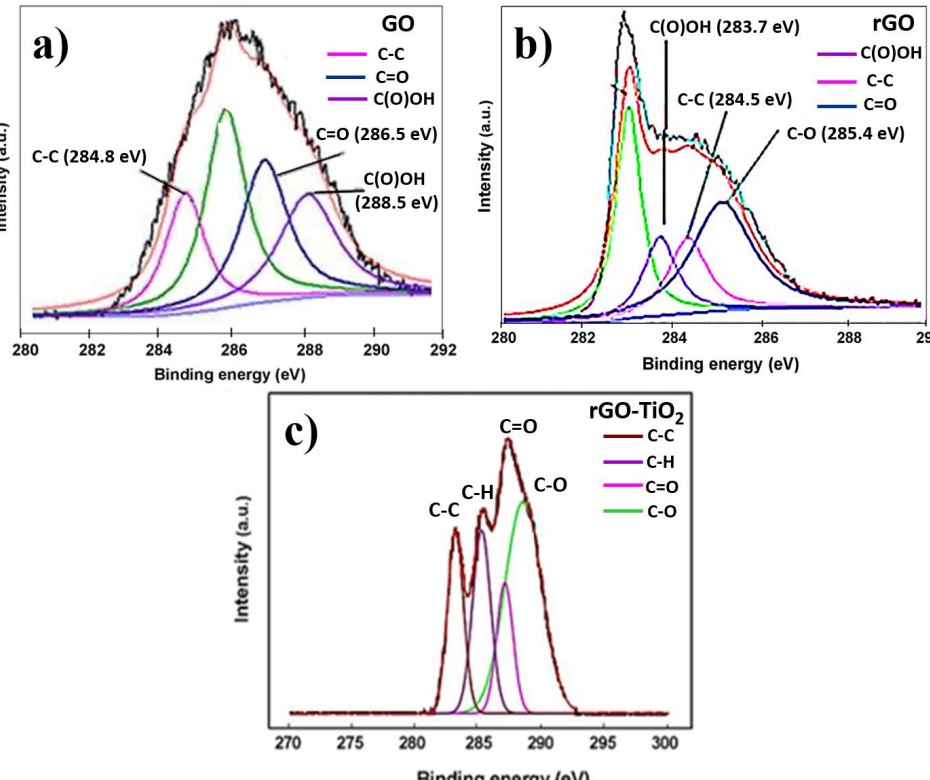

**Figure 4.** XPS binding energy spectra of C1s spectrum of (**a**) GO, (**b**) rGO, and (**c**) rGO/TiO$_2$.

### 3.4. Electrochemical Non-Enzymatic Detection

The electrochemical characteristics of pristine FTO, $TiO_2$/FTO, rGO/FTO, and rGO/ $TiO_2$/FTO samples were investigated using CV scans in a PBS (7.4 pH) solution containing 2 mM D (+) glucose at a scan rate of 50 mV/s and a working area of 1.0 cm$^2$ (Figure 5a). For FTO, it was found that a faradic peak does not exist. It was noticed that rGO/FTO had a clearly defined peak for glucose oxidation that occurred at −0.25 V. At −0.25 V, the $TiO_2$/FTO and rGO/$TiO_2$/FTO oxidation peaks were significantly enhanced. The current rGO/$TiO_2$/FTO reaction was practically double in comparison to the rGO/FTO response when the voltage was set to −0.25 V.

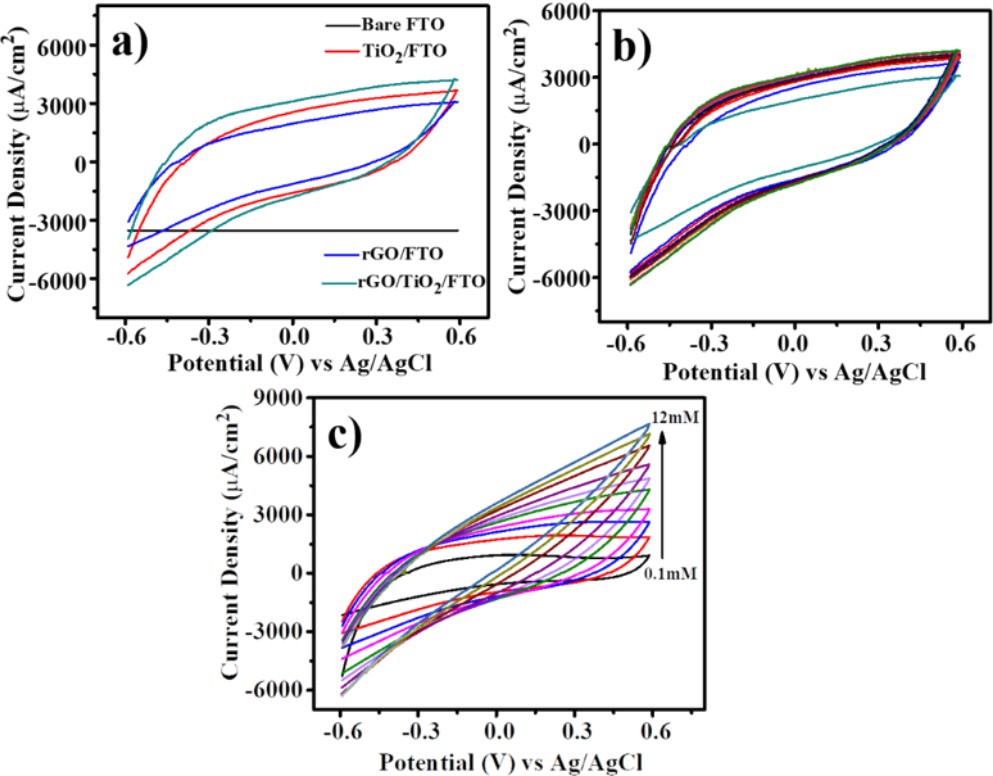

**Figure 5.** (**a**) CV response of various electrodes, (**b**) Effect of variable scan rate on rGO/$TiO_2$/FTO electrode, and (**c**) Effect of variable glucose concentration on rGO/$TiO_2$/FTO electrode.

Consequently, it is confirmed that the rGO/$TiO_2$/FTO composite is essential for obtaining increased sensitivity and electrocatalytic activity. A further investigation into the electrochemical effect on rGO/$TiO_2$/FTO with variable scan rate was carried out using CV which is illustrated in Figure 5b. The electrode was scanned between 10 and 300 mV/s in an electrolyte containing 2 mM D (+) glucose using a CV between −0.6 V and +0.6 V. Regarding current flow, the oxidation that took place at −0.25 V is the most notable. that the rGO/$TiO_2$ nanocomposite undergoes an electrochemical reaction, demonstrating the enzyme detection capability. Furthermore, electrochemical measurements were carried out at the optimal potential of −0.25 V vs. Ag/AgCl, which was optimal.

Figure 5c displays the I-V responses of the modified sensor electrode at a scan rate of 50 mv/s for a variable glucose concentration in the range of 0.1 to 12 mM in a PBS electrolyte solution (7.4 pH) (c). The sensor's current density increases linearly as the glucose concentrations increase. The rGO/$TiO_2$ composite's high electrical conductivity, electron mobility, and electrocatalytic activity are in charge of enhancing the electron transport process between the sensor electrode and the PBS solution during a redox reaction.

### 3.5. Calibration of the Sensor

The calibration curves for various glucose concentration rage are presented in Figure 6. A calibration curve for glucose concentrations in 0.1 to 12 mM is presented in Figure 6a. This curve can be divided into two linear glucose concentration ranges: 0.1–1 mM and 2–12 mM, as presented in Figure 6b,c. For the calibration curve (0.1–1 mM), the equivalent linear regression equations are Ip = 1488.1c + 930.4 and $R^2$ = 0.9862 (N = 4). The sensor's sensitivity determined from the slope of the graph is 1425 $\mu A/cm^2$ (S/N < 3). For the calibration curve (2–12 mM) the equations are Ip = 498.5c + 1503.7 and $R^2$ = 0.9878 (N = 6). The sensor's sensitivity determined from the slope of the graph is 491.75 $\mu A/cm^2$ (S/N < 3).

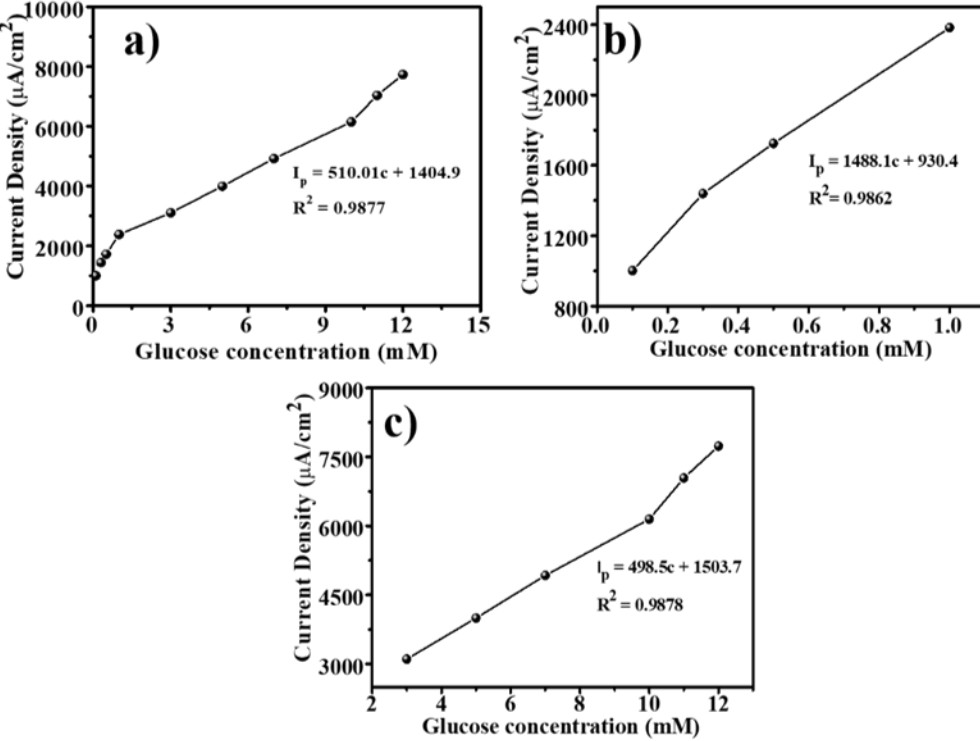

**Figure 6.** Calibration curve for glucose concentration determination using the Glucose liquid color kit (**a**) 0.1–12 mM (**b**) 0.1–1 mM (**c**) 3–12 mM for different glucose concentration.

### 3.6. Selectivity and Stability of the Sensor

The proposed biosensor was examined for interfering species in blood serum, including ascorbic acid, dopamine, fructose, and lactose. The selectivity of the $rGO/TiO_2/FTO$ sensor electrode is shown in Figure 7a. The sample's stability (aging effect) tested periodically for 30 days with a 1 mM glucose concentration, yielding an RSD of 1.14%, is presented in Figure 7b. For ten cycles, the sensor electrode's reproducibility was investigated. To observe the current response of each sensor, 2 mM of glucose concentration was used. The RSD of 1.3% confirms the significant reproducibility.

### 3.7. Repeatability of the Sensor

The repeatability of the proposed sensors was examined by measuring the current response five times over a week. The proposed sensor's performance is compared to previously reported values, and the results are tabulated in Table 2. The sensor exhibits good recovery with an RSD of 1.34%.

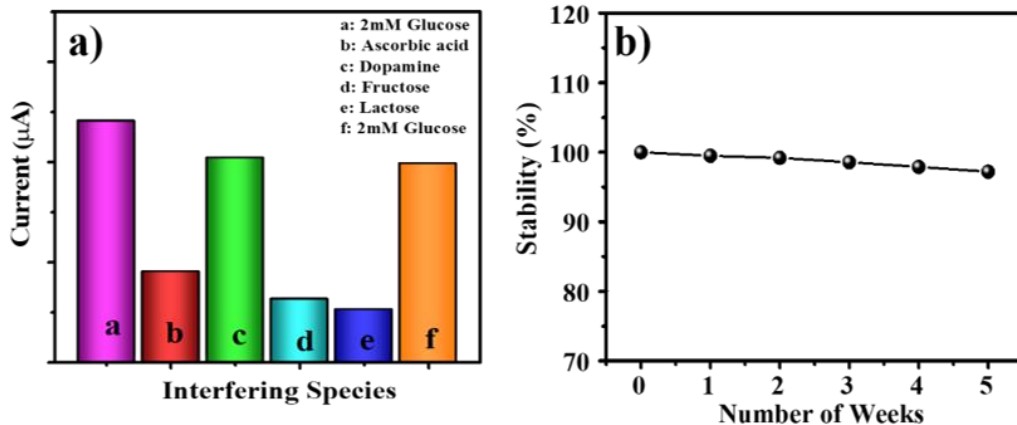

**Figure 7.** (**a**) Selectivity and (**b**) stability of rGO/TiO$_2$/FTO sensor electrode.

**Table 2.** Comparative analysis on sensitivity, detection limit, and linear range of rGO/TiO$_2$/FTO glucose sensor.

| Electrode | Detection Potential (V) | Sensitivity µA/mM1cm2 | Linear Range/mM | Detection Limit/µM | Reference |
|---|---|---|---|---|---|
| rGO/TiO$_2$/FTO | −0.25 | 1425 | 0.1–12 | 0.32 | This work |
| Nafion/GO$_x$/rGO-6/ZnONRs/Au/PET | −0.8 to +0.8 | 2.26 | 11.5 | 37.5 | [42] |
| Ni Ps/ATP/r-GO | 0.0 to −1.5 | 1414.4 µA | 1–710 µM | 0.37 | [43] |
| NiCo$_2$O$_4$-rGO | – | 548.9 | 1–25 | 0.35 | [44] |
| rGO/NiO | −0.6 | 6.2 | 1–15 | 19.35 | [45] |
| GO$_x$/n-TiO$_2$/PANI/GCE | – | 6.31 | 0.02–6 | 18 | [46] |
| 3D Ag-GR-TiO$_2$ | – | 12 | – | – | [47] |
| TiO$_2$–r-GO–PANI | – | – | 10–180 µM | 7.46 | [48] |

*3.8. Investigation of Glucose Level in Blood Serum*

The glucose concentration level in blood serum samples was determined using the prepared sensor electrode. The blood serum samples from 40 to 50 years of age were collected from the certified pathology laboratory. Using the standard dilution method, these samples were multiplied by 100 before being subjected to analysis. The goal of the study was to compare the results for the proposed sensor with the certified values obtained from the pathology laboratory. The experimental data for blood serum glucose concentration are shown in Table 3.

**Table 3.** Investigation of glucose in blood serum and its recovery.

| Serum Sample | Glucose Added (mM) | Glucose Obtained (mM) | Recovery (%) | RSD (%) |
|---|---|---|---|---|
| 1 | 2.00 | 1.97 | 98.5 | 1.5 |
| 2 | 4.00 | 3.92 | 98.8 | 1.2 |
| 3 | 6.00 | 5.87 | 97.5 | 2.2 |
| 4 | 8.00 | 7.89 | 98.6 | 1.4 |
| 5 | 10.00 | 9.71 | 97.1 | 2.9 |

The average recovery rate of 98.2% (RSD 1.88%) exhibits the reliability of the proposed sensor for serum glucose biosensor application.

**4. Conclusions**

The rGO/TiO$_2$ sensor electrode adduced a high sensitivity of 1425 µA/mM cm$^2$ in the linear range of 0.1 mM to 12 mM, which is quite good compared with the available literature data. It shows a quick response time of 5 s, a correlation coefficient (R$^2$) ~0.98, and a LOD of

0.32 μM. It also presents high selectivity towards the interfering species. The RSD of 1.14%, 1.34%, and 1.3% demonstrate excellent stability, repeatability, and reproducibility of the sensor respectively. This reveals the use of rGO/TiO$_2$ electrodes for applications involving glucose sensing. The sensor has an RSD of 1.88% when measuring glucose levels in natural blood serum. Using a commercially available glucose sensor, the measured values agree well with those obtained. Moreover, the process used to prepare rGO and rGO/TiO$_2$ composite is simple, rapid, cost-effective, and eco-friendly which has the potential for commercial/market viability.

**Author Contributions:** Conceptualization, M.G., S.C. and A.G.; Methodology, M.G.; Software, M.U.H.S.; Validation, S.F.S. and R.S.M.; Formal analysis, R.S.M.; Investigation, S.E., R.S.M. and A.G.; Resources, S.F.S.; Writing—original draft, M.G., S.C. and S.E.; Writing—review & editing, R.S.M. and A.G.; Visualization, B.P.; Supervision, S.E., R.S.M. and A.G.; Project administration, S.F.S.; Funding acquisition, S.F.S. All authors have read and agreed to the published version of the manuscript.

**Funding:** The authors sincerely thank the Researchers Supporting Project number (RSP2023R370), King Saud University, Riyadh, Saudi Arabia, for financial support.

**Data Availability Statement:** Not applicable.

**Acknowledgments:** The authors sincerely thank the Researchers Supporting Project number (RSP2023R370), King Saud University, Riyadh, Saudi Arabia, for financial support.

**Conflicts of Interest:** The authors declare no conflict of interest.

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
