# Peer review of "Facile Green Preparation of Reduced Graphene Oxide Using Citrus Limetta-Decorated rGO/TiO2 Nanostructures for Glucose Sensing"

_electronics, doi:10.3390/electronics12020294_

Round 1

Reviewer 1 Report

The paper entitled “Citrus Limetta-Decorated rGO/TiO2 Nanostructures for Enzymeless Glucose Sensing” discussed the electrochemical measurements in the reduced graphene oxide (rGO)/titanium dioxide (TiO2) electrode for glucose sensor. They use the citrus limetta fruit peel waster to reduce the GO to rGO. Then, the rGO/TiO2 composite was obtained using the microwave heating method. In application to sensor, the rGO/TiO2/FTO sensor electrode exhibited a 30 high sensitivity of 1425 µA/mM.cm2 in the linear range of 0.1 mM to 12 mM. After carefully reading the paper, I suggest a major revision with the comments as below:

(1)    How many potions GO can be reduced by aqueous citrus limetta?

(2)    Do author check the ratio of GO over sweet lime peel extract?

(3)    Why choose TiO2 to composite with rGO?

(4)    Please label the GO, rGO and rGO/TiO2 in Fig. 1b

(5)    The author claimed that “The increase in thickness results from the oxygen-containing functional groups being introduced. It can also be noted that the GO sheets were thicker at the edges. Since the oxygen-containing functional groups were primarily combined at the edges of the GO, this is the case.” How to prove this statement?

(6)    In terms of Figure 5, it is very hard to extract the Faradaic peak at -0.25 V, please explain.

Author Response

Response to the Reviewer #1 comments:

------------------------------------------------------------------------------------------------------------

Manuscript ID No.: Electronics-2002789

Title: Facile Green Preparation of Reduced Graphene Oxide Using Citrus Limetta-Decorated rGO/TiO2 Nanostructures for Enzymeless Glucose Sensing Application

Journal: Electronics MDPI

------------------------------------------------------------------------------------------------------------

We are thankful to reviewer for carefully reading our manuscript and commenting on it for publication in MDPI electronics. The comments/suggestions given by reviewer have carefully rectified, and further improvements have incorporated into the revised manuscript. Below are our point-wise responses to the comments/suggestions raised by the reviewer.

The paper entitled “Citrus Limetta-Decorated rGO/TiO2 Nanostructures for Enzymeless Glucose Sensing” discussed the electrochemical measurements in the reduced graphene oxide (rGO)/titanium dioxide (TiO2) electrode for glucose sensor. They use the citrus limetta fruit peel waster to reduce the GO to rGO. Then, the rGO/TiO2 composite was obtained using the microwave heating method. The rGO/TiO2/FTO electrode was applied to the sensor, exhibiting a 30-high sensitivity of 1425 µA/mM.cm2 in the linear range of 0.1 mM to 12 mM. After carefully reading the paper, I suggest a major revision with the comments below:

  • We are thankful to the reviewer for reading the manuscript thoroughly followed a kind recommendation. A point-by-point response has been added in the revised manuscript. The detailed responses are included below and the same changes are made in the revised version of the manuscript. We are sure for accepting this revised manuscript for publication in electronics, MDPI

(1)    How many potions GO can be reduced by aqueous citrus limetta?

  • Thank you for your comment. Approximate reduction of GO by aqueous citrus Limetta is nearly 95%.

(2)    Do the author check the ratio of GO over sweet lime peel extract?

  • Thank you for your comment. Actually, we did not check the exact ratio of GO over sweet lime peel extract as it was not directly related to present work outcome.

(3)    Why choose TiO2 to composite with rGO?

  • TiO2 is used as a catalyst due to its wide surface area, photoelectric activity, high sensitivity, good selectivity, better stability, high electron mobility, and biocompatibility. Though, TiO2 has a good electron transport layer, it may have some limitations such as grain boundary scattering. Noteworthy, enhancement of photocatalytic activity has been seen when rGO is functionalized with TiO2. To facilitate electron transport, rapid enzyme immobilization, and enhanced efficiency, reduced graphene oxide (rGO) is functionalized with TiO

(4)    Please label the GO, rGO, and rGO/TiO2 in Fig. 1b

  • As per the referee’s suggestion, we change GO, rGO, and rGO/TiO2 labelled in Fig. 1b

(5)    The author claimed that “The increase in thickness results from the oxygen-containing functional groups being introduced. It can also be noted that the GO sheets were thicker at the edges. Since the oxygen-containing functional groups were primarily combined at the edges of the GO, this is the case.” How to prove this statement?

  • Thank you for your comment. It is observed by taking the FE-SEM with certain angle. The detail explanation is this regard is given at ttps://doi.org/10.1155/2014/276143

(6)    In terms of Figure 5, it is very hard to extract the Faradaic peak at -0.25 V; please explain.

  • The comment is humbly accepted. But, as per the observations recorded the Faradaic peak was at -0.25V.

Reviewer 2 Report

Review report for 

Facile Green Preparation of Reduced Graphene Oxide Using Citrus Limetta-Decorated rGO/TiO2 Nanostructures for Enzymeless Glucose Sensing

This article presents an eco-friendly method of preparing reduced graphene oxide for the preparation of sensors. There are some issues to solve before the publication.

Abstract part:

  • Maybe TiO2 would be enough for the reader to understand it. It’s better to reduce so many abbreviations in the text (line 20)

  • I would recommend decreasing detailed descriptions in an abstract, like ‘using X-ray Powder Diffraction, Field-emission scanning electron microscopy, Fourier-transform infrared spectroscopy, Raman spectroscopy, and X-ray photoelectron spectroscopy.

  • I would recommend avoiding reference to some literature across the abstract (line 31)

  • Some text repeats the conclusion part and looks hard to understand for readers. So, some precise data like “RSD equals 1.14%, 1.34 and 1.3” - has no sense in the beginning since we don’t know what variables were changed in your experiment (line 34)

  • For the abstract part, I would recommend the author reconsider the structure. Make it more clear and briefer.

Introduction part:

  • I would recommend the author put reference when some sentence is related to some  social phenomena or data (line 46)

  • In line 50 the authors pointed the problem twice. It seems that ‘hypoglycemia’ is a kind of problem, it makes a repeat with the previous part of the sentence ‘To avoid problems

  • It’d be nice to put some references for statements in line 52, line 57, line 67

  • It would make readers confused ‘they limit their application because of their high manufacturing cost ’, who? Make it clear, please. (line 62)

  • Please put an abbreviation expansion when you use the abbreviation for the first time. It’s about GO and rGO - line 71 and  line 90

  • I would recommend separating references. In line 79 we can see 3 ref for one statement. 

  • to overcome the toxic effects of hazardous substances’ includes repetition. line85

  • Seems to repeat, line86

  • Please show the object used to compare then said it is ‘slower reaction rate’. line88

  • I would kindly remind the author to focus on the topic, and remove some general descriptions about this precious fruit like ‘Citrus limetta is globally known for sweet lime, sweet lemon, and sweet limetta. The Indian subcontinent is commonly well known as "Mosambi". It is eaten fresh or in the form of juice…

  • I would recommend authors to pay attention to the logical relationship with the previous sentence. ‘Graphene is functionalized with a metal oxide; it can be used for various applications such as energ….’line 99

  • Please clarify the direct relationship between ‘line106 to 113’ and your research work. 

  • I would recommend making details in the experiment part instead of the introduction part. line123

  • For the whole introduction part, I urgently recommend the author pay attention to the structure, not copy sentences or make several repetitions of the same statement. And also please try to keep the narrative coherent to avoid distracting the reader.

  • Please clarify the aim of the work.

Materials, Results

  • try to avoid well-known abbreviations across the text

  • is it really necessary to list all the ingredients in the text? It confuses the readers since it’s hard to understand their application in the work

  • in section 2.2 - authors claimed that they will produce the GO, but as a result, we can see the collection of rGO. Please try to make it clear.

  • section 2.3 - is it about mg? or what is 15 gm?

  • Some sentences have repetitions from the previous parts of the paper.

  • Fig 1 includes the photo of the lemon. Does sweet lime look like a lemon?

  • How did the authors measure the power of the microwave unit? What is the deviation?

  • The following sections are really hard to follow. 

  • Some figures have low quality

Conclusion:

  • Please avoid copying from your abstract.

  • Check your reference list - it includes some symbols

The paper can not be published in the current view.

Author Response

Response to the Reviewer #2 comments:

------------------------------------------------------------------------------------------------------------

Manuscript ID No.: Electronics-2002789

Title: Facile Green Preparation of Reduced Graphene Oxide Using Citrus Limetta-Decorated rGO/TiO2 Nanostructures for Enzymeless Glucose Sensing

Journal: Electronics MDPI

------------------------------------------------------------------------------------------------------------We are thankful to reviewer 2 for carefully reading our manuscript and commenting on it. The comments/suggestions given by reviewer two are carefully rectified, and improvements are incorporated into the revised manuscript. Below are our point-wise responses to the comments/suggestions raised by the reviewer.

 Abstract part:

  • Maybe TiO2 would be enough for the reader to understand it. It’s better to reduce so many abbreviations in the text (line 20).
  • Thank you so much for this comment, the changes have been made as per the suggestion in the revised version of the manuscript.
  • I would recommend decreasing detailed descriptions in an abstract, like ‘using X-ray Powder Diffraction, Field-emission scanning electron microscopy, Fourier-transform infrared spectroscopy, Raman spectroscopy, and X-ray photoelectron spectroscopy.’
  • As per the suggestion, the detail description of characterization technique is minimized.
  • I would recommend avoiding reference to some literature across the abstract (line 31)
  • Thank you for this suggestion the reference is removed from an abstract.

  • Some text repeats the conclusion part and looks hard to understand for readers. So, some precise data like “RSD equals 1.14%, 1.34 and 1.3” - has no sense in the beginning since we don’t know what variables were changed in your experiment (line 34)
  • Thank you so much for this comment. The statements are revised.
  • For the abstract part, I would recommend the author reconsider the structure. Make it clearer and briefer.
  • I really appreciate you offering this recommendation. The possible changes are made in the revised manuscript.

Introduction part:

  • I would recommend the author put reference when some sentence is related to some social phenomena or data (line 46)
  • Thank for your advice, the line 46 is revised and citied with reference.
  • In line 50 the authors point the problem twice. It seems that ‘hypoglycemia’ is a kind of problem, it makes a repeat with the previous part of the sentence ‘To avoid problems. It’d be nice to put some references for statements in line 52, line 57, line 67
  • Thank for your advice, the lines 42,57 and 67 are revised and citied with reference.
  • It would make readers confused ‘they limit their application because of their high manufacturing cost ’, who? Make it clear, please. (line 62)
  • Thanks for your suggestion. We change: “Though, enzymatic sensors have certain advantages, they limit their application because of their high manufacturing cost”
  • Please put an abbreviation expansion when you use the abbreviation for the first time. It’s about GO and rGO - line 71 and line 90
  • Abbreviation expansion about GO and rGO - line 71 and line 90 are introduced.
  • I would recommend separating references. In line 79 we can see 3 ref for one statement. 
  • Thanks for your suggestion. We change: The astonishing mechanical [9] thermal [10] and electrical [11] graphene properties have captivated tremendous attentiveness.
  • ‘to overcome the toxic effects of hazardous substances’ includes repetition. line85
  • As suggested, we corrected the sentences in the revised manuscript.
  • It seems to repeat line 86
  • Line 85 is revised.
  • Please show the object used to compare, then said it is a ‘slower reaction rate’. line88
  • Thanks for this comment. we change: The reduction of GO can be done using citrus plant extract, which has a slower reaction rate to control the crystal growth
  • I would kindly remind the author to focus on the topic and remove some general descriptions about this precious fruit, like ‘Citrus limetta is globally known for sweet lime, sweet lemon, and sweet limetta. The Indian subcontinent is commonly well known as "Mosambi". It is eaten fresh or in the form of juice.’
  • The statement is revised as per suggestion.
  • I would recommend authors to pay attention to the logical relationship with the previous sentence. ‘Graphene is functionalized with a metal oxide; it can be used for various applications such as energ….’line 99
  • Thanks for this suggestion. We change: Graphene functionalized with a metal oxide; can be used for various applications such as energy storage, electronic applications, sensors, water filters, dye removal, and dye-sensitized solar cells [21–24].
  • Please clarify the direct relationship between ‘line106 to 113’ and your research work. 
  • Thanks, you for this comment. We change: According to these findings, the graphene-based metal/metal oxide nanoparticles had good sensitivity, a wide linear range, and a low detection limit with good selectivity. Hence, rGO/TiO2 is selected in the proposed work. In this work a green synthesis route is used to obtain rGO using phytochemicals in sweet lime peel (citrus limetta) Line 106 to 113 gives the literature review of graphene and its functionalization. This is one of the reasons behind choosing rGO/TiO2
  • I would recommend making details in the experiment part instead of the introduction part. line123
  • As per suggestion, Line 123 is moved to experiment section.
  • For the whole introduction part, I urgently recommend the author pay attention to the structure, not copy sentences or make several repetitions of the same statement. And also, please try to keep the narrative coherent to avoid distracting the reader.
  • As per suggestion, we revised the structure of manuscript and removed repetitions of statements.
  • Please clarify the aim of the work.
  • Thank you so much for this comment. The main aim of this work is to use green synthesized rGO using phytochemicals in sweet lime peel for Enzymeless glucose Sensing application.

Materials, Results

  • try to avoid well-known abbreviations across the text
  • Changes made as per the suggestion in the revised manuscript.
  • is it really necessary to list all the ingredients in the text? It confuses the readers since it’s hard to understand their application in work
  • Changes made as per the suggestion in the revised manuscript.
  • in section 2.2 - the authors claimed that they will produce the GO, but as a result, we can see the collection of rGO. Please try to make it clear.
  • Revision done to make it more clear and understandable
  • Section 2.3 - is it about mg? or what is 15 gm?
  • Yes, it is 15 gm of finely chopped sweet lime peels.
  • Some sentences have repetitions from the previous parts of the paper.
  • As per suggestion we remove repetitions part in revised version of manuscript.
  • Fig 1 includes the photo of the lemon. Does sweet lime look like a lemon?
  • No, It is sweet lime. One can get such sweet limes in local market of India.
  • How did the authors measure the power of the microwave unit? What is the deviation?
  • It was a regular microwave unit having power of 500 W with adjustable timer in the laboratory.

The following sections are really hard to follow. 

  • Some figures have low quality
  • As per suggestion we overall improve the quality of figs.
  • Please avoid copying from your abstract.
  • As per suggestion we modify abstract.
  • Conclusion is updated
  • We updated conclusion part as per reviewer suggestion
  • Check your reference list - it includes some symbols.
  • As per suggestion, we changed the reference and modified it in accordance with the standards format of the electronics journal.

Reviewer 3 Report

The presented paper fulfills the Journal Scopus. This article is based on 48 articles in the literature over the last 10 years. The results are clearly stated. The authors, motivated by cited literature, carried out experiments and confirmed known knowledge. The manuscript presented is interesting but has some shortcomings, including:

(a) how was the binder removal temperature adjusted at the electrode? Does it not affect the state of the composite?

(b) how was the amount of TiO2 in the composite adjusted? When using a different amount of precursor, are the results s

(c) legends missing in Figure 2, curves are not signed, and there are no color explanations?

(d) no data on the XRD base used,

(e) EDS analysis - is the result presented as an average or a score?

(f) the authors suggest that the peak from glucose oxidation is clear, with nothing visible like this on the voltammograms

(g) what is the active surface area of the electrodes, and how does it change relative to TiO2 itself?

(h) how were the calibration measurements performed?

(i) entry 22 in the literature needs improvement in the inventory

Recommendation Regarding This Manuscript: Major revision

Author Response

Response to the Reviewer #3 comments:

------------------------------------------------------------------------------------------------------------

Manuscript ID No.: Electronics-2002789

Title: Facile Green Preparation of Reduced Graphene Oxide Using Citrus Limetta-Decorated rGO/TiO2 Nanostructures for Enzymeless Glucose Sensing Application

Journal: Electronics MDPI

------------------------------------------------------------------------------------------------------------We are thankful to reviewer 3 for carefully reading our manuscript and commenting on it. The comments/suggestions given by reviewer three are carefully rectified, and improvements are incorporated into the revised manuscript. Below are our point-wise responses to the comments/suggestions raised by the reviewer.

The presented paper fulfills the Journal Scopus. This article is based on 48 articles in the literature over the last 10 years. The results are clearly stated. The authors, motivated by cited literature, carried-out experiments and confirmed known knowledge. The manuscript presented is interesting but has some shortcomings, including:

Author reply: We are thankful to the reviewer for reading the manuscript thoroughly and for their kind comments. A point-by-point response has been added to the revised manuscript. The detailed responses are shown below, and a revision has also been made. We hope this manuscript can be accepted for publication after this revision for the sensor journal

(1) how was the binder removal temperature adjusted at the electrode? Does it not affect the state of the composite?

  • Binder removal process was carried out using muffle furnace. The phase of material i.e. TiO2 remained same.
  • (2) how was the amount of TiO2 in the composite adjusted? When using a different amount of precursor, are the results s
  • Thank you so much for this comment. Actually, we have not used much variations.

(3) legends missing in Figure 2, curves are not signed, and there are no color explanations?

  • Thank you so much for this suggestion. We added colure code with name i.e. The D-band of GO (Black color), rGO (Red color), and rGO/TiO2 (Blue color.

(4) no data on the XRD base was used,

  • XRD results were compared with either literature data or JCPDF file for confirming the phase.

(4) EDS analysis - is the result presented as an average or a score?

  • Yes, it is an average score.

(5) the authors suggest that the peak from glucose oxidation is clear, with nothing visible like this on the voltammograms.

  • The comment is humbly accepted and necessary change is made in the statement.

(6) What is the active surface area of the electrodes, and how does it change relative to TiO2 itself?

  • The active surface area of the electrodes is 1 cm2.

(7) how were the calibration measurements performed?

  • Calibration measurements are performed using regression analysis in Microsoft Excel

(8) entry 22 in the literature needs improvement in the inventory

  • As per suggestion Entry 22 is revised

Round 2

Reviewer 1 Report

All questions have been responded very well, and thereby it can be accepted now.

Author Response

  • All questions have been responded very well, and thereby it can be accepted now.
  • We are thankful to reviewer #1 for accept our manuscript and given positive feedback.

Reviewer 2 Report

Review report for 

Facile Green Preparation of Reduced Graphene Oxide Using

Citrus Limetta-decorated rGO/TiO2 Nanostructures for Enzymeless Glucose Sensing Application

This work presents a glucose sensor whose electrodes are made by rGO/TiO2 and its preparation work, among them rGO is obtained by citrus limetta reducing GO. Authors did a lot of work on the paper text, but academic writing is needed to be corrected before the publication.

Abstract Part

Line 24 Please pay attention to the unit.

Line 25 Please clarify the exact variables for relative standard deviation 1.14%, 1.34 and 1.3%

Line 29 I supposed maybe the authors want to write relative standard deviation? If so, please try to fix it, and put the abbreviation ‘RSD’ in line 25, where you firstly mention relative standard deviation.

Introduction Part

Line 59 I recommend authors to be more precise then actuality of the work was revealed. For example, try to avoid phrases like “it has been dubbed "wonder material."”. At least, if it’s the truth, please make a proper reference.

Line 69-73 I recommend moving up, since for readers, it’s much easier to accept the logic which first introduces the shortcomings of traditional methods, then describe your novel green method.

Line 74-77 Could you clarify the relationship of sentence 1 (line 74-75) and 2 (Line 75-77)? They are poorly connected.

Line 80 Does this sentence is a repetition of line 65? If so, please try to avoid repetition.

Line 84 Does this relate to the aim of this work?

Line 88-91 Nice material! But I recommend authors pay attention to the logic of writing, it’s easy to make readers confused, because in the last sentences you introduce the peel composition, then introduce the graphene-functionalized metal oxides without any transitions.

Line 103 Please clarify rGo/TiO2 is selected for what ? As electrodes?

Line 104-106 I recommend the authors put real reason for choosing CV method, I think ‘very little literature on rGO/TiO2-based biosensors’ can not support this choice of method.

Materials and methods

2.1

Line 113 I would suggest the author uniform description of purity, uniform use of ‘(%)’ or ','

Line 117 Please clarify, what is the standard method?

2.2

Line 123 I’m a little confused about this sentence, it looks like a statement, but a process. If authors want to describe the process, I would like to suggest trying to fix this sentence.

2.3

Line 131 I have a question about this process, is it necessary to control the temperature during the whole experiment, because it contains several hours, after that aqueous phytoextract is stored at 4°C. If the experiment temperature is higher than 4°C, does this have an effect on the experimental results? How do authors obtained the stabilized temperature of 4°C? Any deviations? 

Line 136 I would suggest using the exact time instead of ‘overnight’.

2.4

Line 142 Extra full stop is here.

2.5

Questions about whether the oxidized products (author mentioned in line 141 and 142) still effect under temperature 98°C for 8 hours. If they can work in a high temperature for 8 hours, why do authors need to store it in 4°C (line 137)? Why not use normal temperature for storage?

2.7

Line 180 Please check fonts.

3.1

line 228 I recommend putting legend for b) and c) or use some other methods making the curves easy to follow.

Line 229 I recommend reconsider the description, for example we know that a) it’s not X-ray, it’s a result of x-ray testing, etc.

Line 239 Maybe here the author can add something to describe a higher ratio corresponding to higher structural defects.

Line 250 I recommend using ‘is’ instead of ‘=‘.

3.2

Line 274 I recommend that according to these descriptions, mark the corresponding area and point out where it is on the picture 3. It's a little hard for readers to distinguish and find the corresponding position in the picture. 

3.3

Line 311 I think maybe picture 4 were not fully discussed. It seems the authors have experiment data then make a curve fitting, after that making a multiple peak fitting. But it’s easy to confuse readers with curves you didn’t mention in text, but shown in pictures (like green curve).

3.5

Line 348 please clarify the meaning of R2.

(I think for this part is not fully discussed, I didn’t find methodology of calibration…..)

Conclusion part

Line 394 I recommend using RSD directly.

Review in conclusion

I recommend the authors reconsider some sentences in the introduction part, and pay more attention to the logic. The scientific problem is needed to be clarified. Typically the aim of the work presents a separated paragraph - check your target papers. 

Please check all the abbreviations and their extensions, please make sure that you give the abbreviation after extensions which you mentioned it first time.

Please check the space between words and check fonts.

This work contains many processes, but not all the experiment methodology were discussed, like the calibration part.

Author Response

Response to the Reviewer # 2 comments:

----------------------------------------------------------------------------------------

Manuscript ID Number: Electronics-2002789-R2

Title: Facile Green Preparation of Reduced Graphene Oxide Using Citrus Limetta-Decorated rGO/TiO2 Nanostructures for Enzymeless Glucose Sensing Application

Journal: Electronics MDPI

----------------------------------------------------------------------------------------We are thankful to reviewer#2 for carefully reading our manuscript and commenting on it for publication in MDPI electronics. The comments/suggestions given by reviewer have carefully rectified, and further improvements have incorporated into the revised manuscript. Below are our point-wise responses to the comments/suggestions raised by the reviewer.

Abstract Part

  • Lines 24: Please pay attention to the unit.
  • As per the reviewer suggestion the changes have been made in the revised version of manuscript.
  • Line 25: Please clarify the exact variables for relative standard deviation 1.14%, 1.34 and 1.3%
  • Thank you so much for this comment, the necessary clarification has be provided in the revised manuscript.
  • Line 29: I supposed maybe the authors want to write relative standard deviation? If so, please try to fix it, and put the abbreviation ‘RSD’ in line 25, where you firstly mention relative standard deviation.
  • The changes are made as per the suggestions.

Introduction Part

  • Line 59: I recommend authors to be more precise then actuality of the work was revealed. For example, try to avoid phrases like “it has been dubbed "wonder material."”. At least, if it’s the truth, please make a proper reference.
  • The statement has been revised as per your suggestion.
  • Line 69-73: I recommend moving up, since for readers, it’s much easier to accept the logic which first introduces the shortcomings of traditional methods, then describe your novel green method.
  • The order of the statements is changed.
  • Line 74-77: Could you clarify the relationship of sentence 1 (line 74-75) and 2 (Line 75-77)? They are poorly connected.
  • Line 74-75 is moved up to 68-69 for better connectivity.
  • Line 80: Does this sentence is a repetition of line 65? If so, please try to avoid repetition.
  • Yes, it was a repetition hence deleted.
  • Line 84: Does this relate to the aim of this work?
  • The proposed work also aims to obtain best out of waste too. In view of that, fruit peels are used as a green reducing agent to obtain rGO. They are an important source of phytochemicals like flavonoids, pectins, carotenoids, isohesperidin, terpineol alkaloids, essential oils, d-limonene and also beneficial as an antioxidant.
  • Line 88-91: Nice material! But I recommend authors pay attention to the logic of writing, it’s easy to make readers confused, because in the last sentences you introduce the peel composition, then introduce the graphene-functionalized metal oxides without any transitions.
  • Yes, we are agreed for this and as per your comment, the statement is edited in the revised manuscript.

(10) Line 103: Please clarify rGO /TiO2 is selected for what? As electrodes?

  • It is clarified and the necessary changes are made in the manuscript.
  • (11) Line 104-106: I recommend the authors put real reason for choosing CV method, I think ‘very little literature on rGO/TiO2-based biosensors cannot support this choice of method.
  • True. The statement is revised as per your valuable suggestion.

Materials and methods

2.1

(12) Line 113: I would suggest the author uniform description of purity, uniform use of ‘(%)’ or ','

  • The uniform description of purity is maintained in the manuscript as per your comment.

(13) Line 117: Please clarify, what is the standard method?

  • Instead of writing standard method, we have introduced the preparation method in the revised manuscript.

2.2

(14) Line 123: I’m a little confused about this sentence; it looks like a statement, but a process. If authors want to describe the process, I would like to suggest trying to fix this sentence.

  • Yes, statement describes the process.

2.3

(15) Line 131: I have a question about this process, is it necessary to control the temperature during the whole experiment, because it contains several hours, after that aqueous phytoextract is stored at 4°C. If the experiment temperature is higher than 4°C, does this have an effect on the experimental results? How authors do obtain the stabilized temperature of 4°C? Any deviations? 

  • The aqueous phytoextract was stored at 4°C only for 1-2 h in the present case to avoid contamination if there could be any. The extract was brought to room- temperature before use. Hence, there was no deviation in the result.

(16) Line 136: I would suggest using the exact time instead of ‘overnight’.

The exact time is mentioned in the revised manuscript instead of “Overnight”.

2.4

Line 142 Extra full stop is here.

  • All the extra full stops are removed.

2.5

(17) Questions about whether the oxidized products (author mentioned in line 141 and 142) still effect under temperature 98°C for 8 hours. If they can work in a high temperature for 8 hours, why do authors need to store it in 4°C (line 137)? Why not use normal temperature for storage?

  • The aqueous phytoextract was stored at 4°C only for 1-2 h in the present case to avoid contamination. The extract was brought to room temperature before use.

2.7

(18) Line 180: Please check fonts.

  • The fonts are checked and the necessary corrections are made in the revised manuscript.

 3.1

(19) Line 228: I recommend putting legend for b) and c) or use some other methods making the curves easy to follow.

  • The legends are mentioned as per the suggestions given in the previous revision.

(20) Lines 229: I recommend reconsider the description, for example we know that a) it’s not X-ray, it’s a result of x-ray testing, etc.

  • The description is revised as per the suggestion.

(21) Line 239: Maybe here the author can add something to describe a higher ratio corresponding to higher structural defects.

  • The statement is modified in the revised manuscript.

(22) Line 250: I recommend using ‘is’ instead of ‘=‘.

  • The necessary corrections are incorporated.

 3.2

(23) Line 274: I recommend that according to these descriptions, mark the corresponding area and point out where it is on the picture 3. It's a little hard for readers to distinguish and find the corresponding position in the picture. 

  • The statement is modified and the necessary changes are done in the revised manuscript.

3.3

(24) Line 311: I think maybe picture 4 was not fully discussed. It seems the authors have experiment data then make a curve fitting, after that making a multiple peak fitting. But it’s easy to confuse readers with curves you didn’t mention in text, but shown in pictures (like green curve).

  • For the GO sample, the C1s bands can be deconvoluted into three bands attributed to C–C, C=O and C(O)OH bonds at 284.8, 286.5 and 288.5 eV, respectively . The C1s XPS spectrum of rGO shows a significant intensity decrease of C=O and C(O)OH bands in comparison to that of GO, indicating effective deoxygenation of GO after the reduction. However, after being reduced, the peak intensity of oxygen with functionalities was decreased sharply as shown in the figure indicating effective deoxygenation of GO after the reduction. This finding proved the reduction of GO. For the rGO sample, the peaks of C–C, C=O and C(O)OH bands locate at 284.5, 285.4 and 283.7 eV, respectively. Thus, the binding energy values of rGO are shifted. These shifts are assigned to the change in the chemical environment of carbon atoms after reduction. The shifting of  green curve reveals  the removal of oxygen-containing groups of GO using Citrus Limetta.

3.5

(25) Line 348: Please clarify the meaning of R2.

(I think for this part is not fully discussed; I didn’t find methodology of calibration)

  • It is calculated using regression analysis in Microsoft Excel.

Conclusion part

(26) Line 394 I recommend using RSD directly.

  • As per your recommendation the necessary editions are performed in the revised manuscript.

Reviewer 3 Report

Despite the corrections made, the manuscript contains a few errors.

Did the authors not measure the characteristics over a wider current range? Sometimes the peak characteristic of the oxidation-reduction reaction of glucose appears after a much higher potential, which depends on the material used (in most semiconductor materials this is between 0.7 and 0.8 V). Reading values at the 0.6V limit can be subject to significant error.
As for the active area of the electrode, I was not referring to the geometrical dimension-> Randles Sevcik equation
How did electrodes made of pure materials behave in a solution containing no glucose? Is the electrode substrate itself not active to changes in glucose concentration?

Author Response

Response to the Reviewer #3 comments:

----------------------------------------------------------------------------------------

Manuscript ID No.: Electronics-2002789-R2

Title: Facile Green Preparation of Reduced Graphene Oxide Using Citrus Limetta-Decorated rGO/TiO2 Nanostructures for Enzymeless Glucose Sensing

Journal: Electronics MDPI

We are thankful to reviewer 3 for carefully reading our manuscript and commenting on it. The comments/suggestions given by reviewer two are carefully rectified, and improvements are incorporated into the revised manuscript. Below are our point-wise responses to the comments/suggestions raised by the reviewer.

  • Did the authors not measure the characteristics over a wider current range? Sometimes the peak characteristic of the oxidation-reduction reaction of glucose appears after a much higher potential, which depends on the material used (in most semiconductor materials this is between 0.7 and 0.8 V). Reading values at the 0.6V limit can be subject to significant error.
  • The characteristics were not measured over a wider current range. The response of modified sensor electrode was studied at a scan rate of 50 mV/s for different glucose concentrations in the range of 0.1 to 12 mM in a PBS electrolyte solution (7.4 pH). We humbly accept the statement “Reading values at the 0.6V limit can be subject to significant error”, as at higher glucose concentrations the curves are very close to each other.
  • As for the active area of the electrode, I was not referring to the geometrical dimension Randles Sevcik equation
  • The active area of the electrode was not studied using Randles Sevcik equation which will be considered in future research work.
  • How did electrodes made of pure materials behave in a solution containing no glucose? Is the electrode substrate itself not active to changes in glucose concentration?
  • The sensor electrode prepared in the present work is non-enzymatic. Hence, electrode substrate can sense the changes in glucose concentration only after the addition of glucose. Also, electrode behaves like a bare electrode and, without glucose, does not show current variation in a solution.

Round 3

Reviewer 2 Report

The authors have improved the text, but some issues have to be updated before publishing:

1. Fig. 3 is not clear to the readers. For example, authors discribe the thikness of layers, but didn't measure it in the paper. Fig 3 shows just SEM images without measurments of the layers. Try to make it more clear.

2. Figure 4 capture doen't contain the information about the curves in the grapths. Or the legends were absent. 

3. Please check the values for Ip. According to the figure 6 it shows that Current density corresponds to µA/cm^2, but in line 354 and 356 contain µA/mM cm^2 - maybe a missprint?

Author Response

Response to the Reviewer # 2 comments:

------------------------------------------------------------------------------------------------------------

Manuscript ID Number: Electronics-2002789-R3

Title: Facile Green Preparation of Reduced Graphene Oxide Using Citrus Limetta-Decorated rGO/TiO2 Nanostructures for Enzymeless Glucose Sensing Application

Journal: Electronics MDPI

------------------------------------------------------------------------------------------------------------

We are thankful to reviewer #2 for carefully reading our manuscript and commenting on it for publication in MDPI electronics. The comments/suggestions given by the reviewer have been carefully rectified, and further improvements have been incorporated into the revised manuscript. Below are our point-wise responses to the comments/suggestions raised by the reviewer.

Comment 1: Fig. 3 is not clear to the readers. For example, the authors describe the thickness of layers, but didn't measure it in the paper. Fig 3 shows just SEM images without measurements of the layers. Try to make it more clear.

Author reply: Thank you very much for the valuable comment. We have corrected the lines in the main body text about the thickness of GO sheet and the related reference has been quoted for your kind reference. 

As per the reported literature [38], the thickness of GO sheet was significantly greater than the thickness of the single-layer graphene. The increase in thickness results from the oxygen-containing functional groups that were introduced. It can also be noted that the GO sheets were thicker at the edges [38].

Comment 2: Figure 4 capture doesn’t contain the information about the curves in the graphs. Or the legends were absent. 

Author reply: Thank you very much for the worthy comment. We have added the information about the curve in the graph and in text as well for better understanding and also added the legends in the graphs.

Comment 3: Please check the values for Ip. According to the figure 6 it shows that Current density corresponds to µA/cm^2, but in line 354 and 356 contain µA/mM cm^2 - maybe a misprint?

Author reply: Thank you so much for pointing out of typo error, we have rectified the sentence with the proper unit i.e. µA/cm2.
